# An Advanced Rider-Cornering-Assistance System for PTW Vehicles Developed Using ML KNN Method

**DOI:** 10.3390/s23031540

**Published:** 2023-01-31

**Authors:** Fakhreddine Jalti, Bekkay Hajji, Alberto Acri, Michele Calì

**Affiliations:** 1Laboratory of Renewable Energy, Embedded System and Information Processing, National School of Applied Sciences, Mohammed First University, Oujda 60000, Morocco; 2Department of Engineering, University of Messina, 98158 Messina, Italy; 3Electric, Electronics and Computer Engineering Department, University of Catania, 95125 Catania, Italy

**Keywords:** powered two-wheeler dynamic behavior, maximum cornering velocity, advanced rider assistance systems, k-nearest neighbor, machine learning

## Abstract

The dynamic behavior of a Powered Two-Wheeler (PTW) is much more complicated than that of a car, which is due to the strong coupling between the longitudinal and lateral dynamics produced by the large roll angles. This makes the analysis of the dynamics, and therefore the design and synthesis of the controller, particularly complex and difficult. In relation to assistance in dangerous situations, several recent manuscripts have suggested devices with limitations of cornering velocity by proposing restrictive models. However, these models can lead to repulsion by the users of PTW vehicles, significantly limiting vehicle performance. In the present work, the authors developed an Advanced Rider-cornering Assistance System (ARAS) based on the skills learned by riders running across curvilinear trajectories using Artificial Intelligence (AI) and Neural Network (NN) techniques. New algorithms that allow the value of velocity to be estimated by prediction accuracy of up to 99.06% were developed using the K-Nearest Neighbor (KNN) Machine Learning (ML) technique.

## 1. Introduction

Thanks to the introduction of recent technologies, systems and devices that interpret signals from various sensors and computer-controlled components and safety standards have reached high levels of reliability both for the driver and for the passengers of two-wheeled vehicles [1,2]. Unlike what happened for four-wheeled vehicles, the progress and technological evolution of safety devices for two-wheeled vehicles began with two decades of delay. Throughout the years, improvements in relation to safety have concerned almost exclusively the passive features: airbags, protective clothing, helmets, etc. For example, if the ESC has been available on cars since 1995, the Kronreif und Trunkenpolz Mattighofen (KTM) Austrian motorcycle company introduced the Motorcycle Stability Control (MSC) to a motorcycle only in 2013.

Although model-free design methods have been investigated considerably, up to now, most control synthesis methods and tools are categorized as physic model-based, and the system subject to control is evaluated as a type of dynamic model of the system itself.

The design of the control system requires accurate mathematical modeling since the closed-loop performance is strictly influenced by the dynamic behavior of the system and the use of very accurate sensors and measuring instruments. The control system is characterized by many closely interacting subsystems; advanced control systems are requested for a competitive performance, and an explicit mathematical model is thought for their design; the system seems to be safety critical and an extensive validation of the closed-loop stability and performance by simulation is considered essential.

### Related Work

The importance of mathematical models is universally recognized for the design of the control of these safety systems. For this purpose, different types of mathematical models are used, such as detailed models for dynamic system simulation in multibody environments and numerical models based on AI algorithms for control design.

Mathematical models dealing with dynamic simulation are generally quite heavier [3,4,5]; linearization is not always practicable and often involves a loss of accuracy [6,7]. This involves the use of an overpowered on-board computer to assist the driver in better confronting the risks. This is, of course, far from expedient for the motorcycle industry; for this reason, other research was prompted to arise with the aim of simplifying the model [7] or estimating the parameters [8]. Additionally, the two-wheeled is a vehicle whose balance is difficult to recover; for example, there are few or even no actuators available to correct the slip on the bend.

In their previous works, the authors tried to evaluate the maximum cornering velocity using detailed multibody models attempting to also consider the influence of the pilot’s movements [9,10,11,12,13]. Indeed, two vehicles, even if they have the same characteristics (weight, wheelbase, tires, geometry, etc.), can perform very different dynamic cornering behaviors depending on the rider that maneuvers the vehicle. Below in Table 1, the main control systems designed and/or presented in the scientific literature have been classified, indicating for each of them the principle, the use of the instrumentations and the advantages and disadvantages.

A driver who knows his vehicle well has developed an aptitude so to maneuver it on a bend better than the most intelligent existing robot can do (autonomous motorcycles [19]). Our goal is to strengthen the driver’s skill by helping him know to bend more appropriately and providing other information that he could not get directly with sensors.

To achieve this goal, we developed an efficient model based on ML for the prevention of PTW cornering risks with minimum error. The ML techniques reported in the literature are: Multiple Linear Regression (MLR), the Decision Tree (DT), Artificial Neural Network (ANN) and K Nearest Neighbor (KNN), which are used in regression or classification contexts. Our choice was addressed to the ANN and KNN regression approaches because of their suitability for the prediction of continuous quantities and given our previous comparison of effectiveness [20]. Therefore, the choice to use the KNN method is for the following three reasons:As explained in Section 3.1 (description of the methodology), given the nature of the outputs (numerical observations), the form of regression of KNN appears to be the most suitable.We conducted a comparison of the different methods and found that the ANN and KNN gave the best results, but the ANN was unstable given its probabilistic behavior. By calculating R^2^ indicator, we found that the ANN algorithm gives different values that deviate from 25% to 90%.It is optimal, uses less resources and gives good performance, simple to deploy and to interpret.

The challenges this study aims to address are:


The complexity of the PTW (Powered Two Wheelers) dynamics;The lack of driver assistance systems for this type of vehicle;The real-time dynamic parameters sharing of information;The lack of databases in this field.


These challenges are the main obstacles that obstruct the constitution of a development framework based on experience feedback and the design of an affordable driver assistance system for the majority of users of this type of vehicle.

The majority of studies that addressed this topic (curve assistance systems) used an instrumentational approach based on sensors or automatic estimators, but we could cite previous studies conducted in the same spirit [21] in which the author aimed to characterize critical curve situations using AI classification algorithms. Indeed, our work complements his study.

On the other hand, at this stage of the state of the art, the framework has not yet been completely finalized, and the works that arise complement each other. Notably, the use of image recognition and AI for visual detection by camera, which is the main field currently analyzed by researchers.

The research work carried out in this manuscript is organized as follows. Section 2 illustrates the proposed approach used to develop the rider-cornering-assistance system; the formulation, the dynamic quantities considered, and their variation ranges are shown. Section 3 describes the evaluation of the V_curve_ correction coefficients through ML based on the KNN regression approach. Section 4 reports the results interpretation and discussion. Finally, in Section 5, the conclusions are showed.

## 2. Cornering Assistance System Formulation

The system studied consists of a triplet (vehicle, trajectory, driver}; they are related to each other quantities that were null during the longitudinal trajectory (angles, forces and moments, etc.). The system succeeded in going through the curve if the initial speed to take the bend was correctly estimated and if, during the bend, the balance was not disturbed by the environment and/or the driver maneuvers [9].

Efficient algorithms based on ML for the prevention of the PTW cornering risks with a minimum error were developed. ANN and KNN regression approaches were used to predict with high accuracy the continuous quantities involved in the cornering dynamic of the PTW vehicle. A dataset was built by utilizing data from an open dataset source (DataMC.org—Motorcycle Data Acquisition&motorcyclespecs.co.za) that were exploited using known dynamic models and previously developed multibody dynamic models [10]. Our models were able to consider the rider’s movements and evaluate the dynamic response of the PTW vehicle.

The method developed was based on an evaluation of the sideslip angle. During curvilinear trajectory, each vehicle had its characteristic dynamic forces, and these depended on the vehicle geometric parameters, the contact road-tire and the rider actions (body movements). In Figure 1, the dynamic forces involved in the vehicle equilibrium during the curvilinear trajectory with non-zero sideslip angles are shown.

PTW vehicle curvilinear trajectory was controlled by the handlebar, which was connected to the front forks; unlike a car, simply steering does not produce an effective turn; the riders are required to shift their weight and lean the vehicle to some angles (dependent on the speed and turn radius) to maintain a correct trajectory. As with all circular motions, the resultant between lateral force and rolling resistance in the front tire and lateral force and driving force in the rear tire is a centripetal force that pulls the PTW vehicle into the center of the turn. The tires provide the needed friction to maintain contact with the road and the necessary front and rear sideslip angles [22].

For each PTW vehicle dataset “i”, the equilibrium control quantities (reported in Figure 1) are presented as a function of other parameters that authors considered as determinant parameters based on the results of their dynamic simulations. Next are reported, divided by categories, these parameters used to develop effective algorithms based on ML for the prevention of PTW cornering risks.

### 2.1. Vehicle Parameters

#### 2.1.1. Know Parameters

In Figure 2, the PTW main geometrical and inertial parameters used in the multibody approach are shown:
PTW geometry: *p*: wheelbase; a: trial; ε: caster angle;Front assembly: *M_f_*: front mass;Rear assembly: *M_r_*: rear mass;Front wheel: *R_f_*: wheel radius;Rear wheel: *R_r_*: wheel radius.

One simple and common way to characterize a PTW vehicle is by the ratio:(1)$$R_{n}=\left(a_{n}/b_{n}\right)\;\left(M_{f}/M_{r}\right)$$

#### 2.1.2. Unknown Parameters

In motion, some parameters changed their values continuously, especially in cornering. These parameters participated in calculating the different forces and moments that determined the equilibrium. We can mention in particular the coordinates of barycenter *G_f_*, the coordinates of barycenter *G_r_*, the steering damper *c*, *K_λf_* cornering stiffness, *K_ϕ__f_* the camber stiffness, *K_λr_* cornering stiffness, *K_ϕr_* the camber stiffness.

### 2.2. Tire-Road Parameters

The analysis of the relationship between tires and the road is the subject of extensive studies [23,24]. Our study mainly focused on lateral force equilibrium conditions. In Figure 3, the scheme of the lateral balance of the PTW vehicle during the curvilinear trajectory is shown [10,23,24].

The wheel–road contact is manifested by a deformation of the tire contact surface. The resulting shape was a function of several parameters, in particular the characteristics of the vehicle (tire pressure, load, etc.), the physical quantities generated by the cornering action (roll angle, sideslip angle, etc.) and any presence of lateral forces and braking or driving torques introduced further deformations to the contact patch. With respect to the longitudinal “x” and transversal “y” axes, the patch was not usually symmetrical.

It follows that within the framework of a predictive approach to estimate the said parameters, we rather sought the indicators that could differentiate one vehicle from another. We selected the following parameters:The tires wear as a function of mileage traveled;The adherence to the road;Tire stiffness, a combination of tire pressure and tire type.

### 2.3. Maneuver Parameters

Despite many studies carried out to model the rider’s behavior [10,25,26], motorcycle dynamics still represent a notoriously thorny topic to deal with, mainly due to the following problems:Because of the presence of the steering head, the description of vehicle kinematics will be complex;To ensure the control of motorcycles, driver action is always required, and this is a variation of driving conditions, such as speed;Riding style, such as the rider’s skill and experience, greatly affects vehicle performance.Figure 4 shows the scheme of PTW driver control system adopted in this study.

In the present study, the PTW maneuverability condition was expressed as a function of two parameters:The type of vehicle: two vehicles that are the same, or very similar, have a higher correlation than motorcycles with different characteristics;The driver: drivers are always different since they may travel the same route at different times and mileage.

### 2.4. Resultant of Parameters

Following the list of different parameters influencing the dynamics of the PTW vehicle on curves, we can state the complexity of quantifying some parameters or even clarifying a correlation between the parameters, knowing that most of them evolve with time. Indeed, several works have tried to estimate dynamic parameters using conventional methods. In particular, multibody models are widely used to estimate roll angle [27], lateral dynamics forces [28], and attitude estimation [29,30].

The authors’ idea was to estimate an appropriate value of the heeling angle of PTW vehicles by developing estimation algorithms based on on-board measurements. This is mainly due to the fact that the sensors available to measure this variable turn out to be bulky and expensive. Currently, algorithms are used for four-wheeled vehicles; however, for two-wheeled vehicles, it is still an open topic. In the context of two-wheeled vehicles, the authors proposed neural network estimation algorithms that consider the standard on-board measurements available in modern motorcycles to study the role of the most significant signals for estimation.

The approach used in this paper served as a preliminary analysis of this estimation problem because it did not require the direct derivation of physical parameters of motorcycle dynamics in its application. The experimental data collected covered a rich number of maneuvers (66 tests) and were used to optimize the developed algorithms, as many maneuvers were used to analyze the effectiveness of the algorithms.

Thus, our approach suggested an estimation of the parameters of the system considering all the factors that play a role in lateral dynamics (Equation (2)):(2)$$V_{curve}=C_{Veh}\cdot C_{ct}\cdot C_{Dr}\cdot V_{ref}$$
where the coefficients are functions of the parameters described in Table 2:*C_Veh_*: Correction coefficient linked to the characteristics of the vehicle; this one is expressed as:
(3)$$C_{Veh}=f\left(R_{n},\;R_{f},R_{r}\right)$$

*C_ct_*: Correction coefficient linked to the road-tire relationship; it is expressed as:


(4)
$$C_{ct}=f\left(Ml_{Tire},\mu_{lat},P_{Tire},Tire_{Type}\right)$$


*C_Dr_*: Correction coefficient linked to the driver’s behavior; it is expressed as:


(5)
$$C_{Dr}=f\left(C_{Veh},\;C_{ct},\;Time,\;Ml_{driver}\right)$$


*V_ref_*: Velocity of the vehicle on the curve $c_{position}$ performed by a referent driver at Temperature “*T*”, Humidity “*H*”, and visibility “*Vis*”, expressed finally by:(6)$$V_{ref}=f\left(c_{position},T,H,Vis\right)$$

In case of the referent vehicle {Driver, Vehicle}, the velocity will only depend on the driver records on similar conditions of the environment (correction coefficient = 1):(7)$$V_{curve}=V_{ref}\left(c_{position},T,H,Vis\right)$$

## 3. Evaluation of the V_curve_ Correction Coefficients through the ML KNN Method

### 3.1. Methodology Description

Table 2 classifies the previously introduced parameters, considering their nature and the proposed method of estimation or measurement. Table 3 reports the values of the parameters used in the first 10 of the 66 tests carried out. The complete table with all the parameters used in the 66 tests is shown in Appendix A.

Our approach was based on credible learning, which means that in the learning phase, we built the dataset considering only referent vehicles equipped with roll angle sensors, accelerometer and drivers having a performant record and riding the most performant vehicle. All referent vehicles travelled the same training trajectory. In addition to the usual equipment, such as the tachometer, each referent vehicle was equipped with the following sensors:Direct TPMS for measuring the pressure and temperature of each tire;Infrared sensor for detecting weather and light conditions;GPS system (possibility of connection with other equipment, such as smartphones);Electro-optical sensor combined Silicon Sensing MEMS (Micro Electro-Mechanical System) gyroscope (CRS-07) for roll angle detection.

Thus, in the training phase and in each study case, between 291 and 300 experiences covered different situations in the appropriate range visible in the last column present in Table 2. The values reported in the section on the training camps varied according to the parameters of the categories analyzed.

Considering the nature of outputs (parameters), we chose regression ML techniques using the K nearest neighbor (KNN). We used “feature similarity” to predict the values of any new data point [31]. This means that a value was assigned to the new point based on how it resembled the points in the training set. The output result was evaluated using Equation (8).
(8)$$\tilde{y}\left(\tilde{x}\right)=\sum_{i=1}^{k}w_{i}y\left(x_{i}\right)$$
where *w_i_* is the weighting factor for *x_i_*, *k* is the number of neighbors, *y* is the response variable and $\tilde{y}$ is the response variable prediction.

### 3.2. Friction Coefficient (μ_lat_) Estimation Approach

As we can see in Table 2, parameters are either an unchanged quantity or a measurement that evolves over time. The parameter that resides unknown is friction.

In our previous paper [31], we managed to estimate the adherence value thanks to an approach based on the ML regression method KNN. This method is true only if the vehicles covered by the dataset are similar with the same characteristics and operate under the same conditions. Thus, the database was constructed based on the following criteria for each measure $x_{i}$:
The quantities {*R_n_*, *R_f_*, *R_r_*, *Tire_Type_*} are assumed as provided;The quantities {*Ml_Tire_*, *T*, *H*, *Vis*, *P_Tire_*} are located in an interval at ±10%.

Consequently, based on the precited conditions for the dataset, the regression variables are expressed as follows: $x_{i}=\left\{V_{i}{}_{réf},\;\phi_{r},\;c_{position}\right\}$, *k* = 2 number of neighbors, $y=\mu_{i}{}_{lat}$ which is the response variable and $\tilde{y}$ is the response variable predicted. On the training phase, lateral adherence was calculated by using Equation (9) [9]:(9) μlat=ρg·Vx2−ϕr1−ρg·ϕr·Vx2
where *µ_lat_*, *ρ*, *v_x_*, ϕ*_r_*, *_g_* are respectively: coefficient of static friction, the curvature of the turn, the longitudinal speed, the tilt angle of the road, and gravity.

On the operating phase, for a given curvature c, based on the measures of *V_ref_* and ϕ*_r_* we evaluated directly the adherence of this section of the road by our function $\tilde{y}\left(\tilde{x}\right)$. In Figure 5, the comparison between *μ_lat_* coefficients predicted and measured is shown.

The similarity between predicted values and real values allowed us to repeat the learning analyzed in [31], so that the new algorithm included the theoretical values. As for the original forecast, the main results are shown below.

Considering that the prediction of the adherence coefficient was the subject of our previous paper. We meant to repeat the learning so that the new algorithm took into account the theoretical values. Hence, the similarity between the predicted and real values.

Likewise, the original prediction here is the results.

As an output, the road grip was evaluated using test data as a function of velocity (Input). In Figure 6, we compared the prediction results given by the two techniques (KNN 
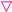
 and ANN *) with the experimental data (○).

We found that the predicted road grip provided by the KNN model was closest to the experimental values. More than that, ANN was unstable considering its probabilistic behavior. By calculating R^2^ indicator, we found that the ANN algorithm gave different values that varied from 25% to 90%.

### 3.3. Vehicle Characteristic Estimation

To evaluate the value of *C_veh_*, we built another dataset of referent vehicles but with different criteria for each measure *x_i_*:
The quantities {*R_n_*, *R_f_*, *R_r_*} are variable;The quantities {*Tire_Type_*, *c_position_*} are assumed as provided;The quantities {*Ml_Tire_*, *T*, *H*, *P_Tire_*, *μ_lat_*} situated within an interval with mean values ±10%.

Thus: $C_{ct}\approx C_{Driver}\approx 1$

Then, based on the above conditions for the dataset, the regression variables were expressed as follows: $x_{i}={\left\{R_{n},\;R_{f},\;R_{r}\right\}}_{i}$, k=2 number of neighbors. The measured correction coefficient of the vehicle $y=C_{Veh_{i}}=V_{i}/V_{ref}\left(c_{i},\;R_{n},\;R_{f},\;R_{r}\right)$ which is the response variable and $\tilde{y}$ is the response variable prediction.

In the operating phase, for a given vehicle and based on the characteristics $\left\{R_{n},\;R_{f},\;R_{r}\right\}$ we evaluated directly the coefficient *C_Veh_* by the function $\tilde{y}\left(\tilde{x}\right)$. In Figure 7, the comparison between *C_Veh_* coefficients predicted and measured is shown.

### 3.4. Contact Road-Tire Prediction

To create a function that predicts the contact road-tire relationship, we built another dataset of the referent vehicle but with different criteria for each measure $x_{i}$:
The quantities {*Tire_Type_*, *P_Tire_*, *μ_lat_*, *Ml_Tire_*} are variable;The quantities {*R_n_*, *R_f_*, *R_r_*, *c_position_*} are assumed as provided;The quantities {*T*, *H*, *Vis*} situated within an interval with mean values ±10%.

Thus: $C_{Veh}\approx C_{Driver}\approx 1$

Then, based on the precited conditions for the dataset, the regression variables are expressed as follows: $x_{i}={\left\{Tire_{Type},\;P_{Tire},\;\;\mu_{lat},\;Ml_{Tire}\right\}}_{i}$, k=2 number of neighbors, the measured correction coefficient of the vehicle $y=C_{ct_{i}}={\frac{V_{i}}{V}}_{ref}\left(c_{i},\;Tire_{Type},\;P_{Tire},\;\mu_{lat},\;Ml_{Tire}\right)$, which is the response variable and $\tilde{y}$ is the response variable prediction.

In the operating phase, for a given vehicle and based on the characteristics {*Tire_Type_*, *P_Tire_*, *μ_lat_*, *Ml_Tire_*} we evaluated *C_ct_* coefficient using $\tilde{y}\left(\tilde{x}\right)$ function. In Figure 8, the comparison between *C_ct_* coefficients predicted and measured is shown.

### 3.5. Rider Behavior Prediction

This time, to predict rider behavior, we built a dataset with ordinary vehicles for which all the other parameters were fixed except those that we assigned to the driver, namely: Time and the record of the driver mileage *Ml_driver_*. Thus:$C_{Veh}\approx C_{Ct}\approx 1$.

Then, based on the precited conditions for the dataset, the regression variables were expressed as follows: $x_{i}={\left\{Time,\;Ml_{driver}\right\}}_{i}$,k=2 number of neighbors, the measured correction coefficient of the vehicle $y=C_{Driver_{i}}={\frac{V_{i}}{V}}_{ref}\left(c_{i}\right)$ which is the response variable and $\tilde{y}$ is the response variable prediction.

In the operating phase, for a given rider and based on the data {*Time*, *Ml_driver_*}, we can evaluate *C_Dr_* coefficient using $\tilde{y}\left(\tilde{x}\right)$ function. In Figure 9, the comparison between *C_Dr_* coefficients predicted and measured is shown.

### 3.6. Overall Results

In this section, the impact of the different correction coefficients on the accuracy of the maximum predicted cornering velocity (*V_max_*) calculation is shown. Figure 10 shows a signal enveloped within a frame (brown line) that follows the referent velocity realized by the referent vehicle in 66 different tests. The correlation attests to the effectiveness of the prediction method followed. Please note that the trajectory is the same, but the other conditions are different for the 66 different dynamic situations tested.

Comparing the predicted velocity (dotted line) with the real velocity (continous line) of each scenario that considers the application of the correction coefficients (*C_Veh_*, *C_ct_*, *C_Dr_*), we can observe the high precision of the model. To evaluate the accuracy, we relied on the measurement of MAE (mean absolute error) using Equation (10):(10)$$MAE=\frac{1}{n}\sum_{i=1}^{n}\left|y_{i}-\hat{y_{i}}\right|$$
where *n*, *y_i_*, $\hat{y_{i}}$, are respectively: the number of measurements; the value of measurements; the corresponding predicted value; the mean of the measurements. The results shown in Figure 11 highlight a relatively stable prediction for different values of K, but better accuracy is given by K = 2 for each algorithm.

However, different levels of performance were observed for each coefficient, which should be the subject of several factors discussed in the following section.

In fact, the performance of the algorithms does not behave the same way with the variation of K (see also Figure 10). Nevertheless, the only case that reacts significantly to the variation of K is that of *C_ct_* prediction. This is explained by our database layout: the KNN algorithm exploits the correlation between the values; thus, the results are limited to the extent of the database that we have used and the diversity of the scenarios that it represents.

## 4. Discussion

Using the KNN-ML technique, a prediction model of the lateral dynamics of PTW vehicles was developed. The vector of inputs used in the model included the main parameters that characterize the system (vehicle, driver, environment). This choice is not exhaustive, but the acquisition of more measurements requires the possibility and verifiability of ensuring reliable learning.

The developed prediction model allowed the evaluation of Vmax by means of correction coefficients in curvilinear trajectories to implement an effective ARAS. The correction coefficients introduced in the previous section showed a different accuracy. Below, the interpretation of the results is reported for each correction coefficient:Vehicle characteristics estimation *C_veh_*: The results showed a stability of the prediction with good accuracy (MAE = 2.9% in the testing phase); this is mainly due to the fact that inputs are known data (excluding tires) that are objectively identified;Contact road-tire prediction: This is a complex function. In fact, having neglected the *K_λ_* stiffness coefficients for the front and rear assemblies, which are notably impacted by the effect of the shock absorbers, biased the results. On the other hand, given the modest amount of data on our Dataset (291 measurements in total), the algorithm could not obtain sufficiently efficient results (MAE ≈ 35%) despite acting on the coefficient K (Number neighboring points). Consequently, we counted on the application on a more consistent scale to be able to collect a sufficient quantity of data to improve this ratio;Rider behavior prediction: We obtained excellent results of precision and stability (MAE ≈ 0.94%). In fact, the simplicity of the parameters made it possible because we chose to reflect only two aspects that could characterize the driver (mileage, time).

The implementation of a conventional method involved the use of a considerable number of sensors; this kind of equipment was not affordable for every user. As for our developed model, the equipment needed was restricted to data sharing and continuous improvement of the algorithms.

The effectiveness of the developed method was also proved by comparing the maximum cornering velocity values obtained with the present algorithms and the same values calculated using the parametric multibody model of the PTW vehicle and rider developed by one of the authors in the work “An advanced multibody model for evaluating rider’s influence on motorcycle dynamics” [10].

By entering the values of the geometric and inertial parameters used in the evaluation of the above algorithms, it was possible to have a numerical validation of the results obtained using the multibody model described in [10]. The results obtained are shown in Figure 12.

The maximum values of cornering velocity V_mb_ calculated using the parametric multibody model of the PTW vehicle had an average difference from the same values evaluated with the algorithms developed in the present work V_ref_ always less than 18%.

No AI-based project that proposed an ARAS-PTW was found in the literature. Only the work [32,33] presents similarities to the present research. Here, the authors proposed an approach based on a mathematical estimate. The function introduced is as follows:(11)$$J\left(x,u\right)=W_{a}\left(a_{x},u\omega_{\Psi }\right)+W_{j}\left(j_{x}\right)+W_{j}\left(j_{\Psi }\right)-u_{x}$$
where *W_a_* is the acceleration envelope function and *W_j_* is the jerk envelopment function. An additional term was introduced to promote speed.

If we compare the proposed Equation (11) with our proposed one Equation (2), we can state considering the difference of approaches that our approach considers all the variables (Environment, Vehicle, rider) and is based on learning and the accumulation of experiences, while the study proposed by [32] presents a real-time approach that requires access to different parameters (sensor layout).

As shown in Figure 13, the author of [32] supposes that the initial speed to start the curve is already known (velocity in point A); this constitutes the main difference from our research. Precisely, this data is unknown since we cannot have an advanced knowledge of the curve started. This is where our solution comes from, which uses historical data to provide advanced knowledge of the curve to be started.

Similar to our previous article based on AI [33], the present research aims to offer assistance to the rider in critical situations by the acquisition of unknown parameters using ML. The goal of the present research was to be able to continuously increase driving skills while taking advantage of the magic of AI to prevent critical situations.

## 5. Conclusions

The work carried out aimed to set up a framework for an ARAS-PTW by using machine learning to predict values that could not be measurable or for which instrumentation may be expensive.

Thus, we chose to analyze cornering since it is one of the most critical dynamic situations that PTW vehicles can address. Our approach consisted of considering coefficients that must be deployed to link a PTW vehicle (user of the solution) to a reference vehicle whose dynamic parameters are correctly identified and that is placed under our total supervision.

The resulting algorithms were satisfactory given that the initial data that we used were only experimental data with a limited number of records, and for *C_ct_* (correction coefficient linked to road-tire interaction) correction, we did not consider one parameter that may standardize the measures. In fact, the stiffness of the front and rear assemblies should be integrated in future works.

The accuracy with which we measured using the proposed correction coefficients could be (in worst scenario) 38.83%, but the new algorithms can allow us to estimate the value of velocity by a prediction accuracy of up to 99.06%.

Our model was meant to be developed in further steps in the frame established in our research work, but it could also be used as a framework for other works.

## Figures and Tables

**Figure 1 sensors-23-01540-f001:**
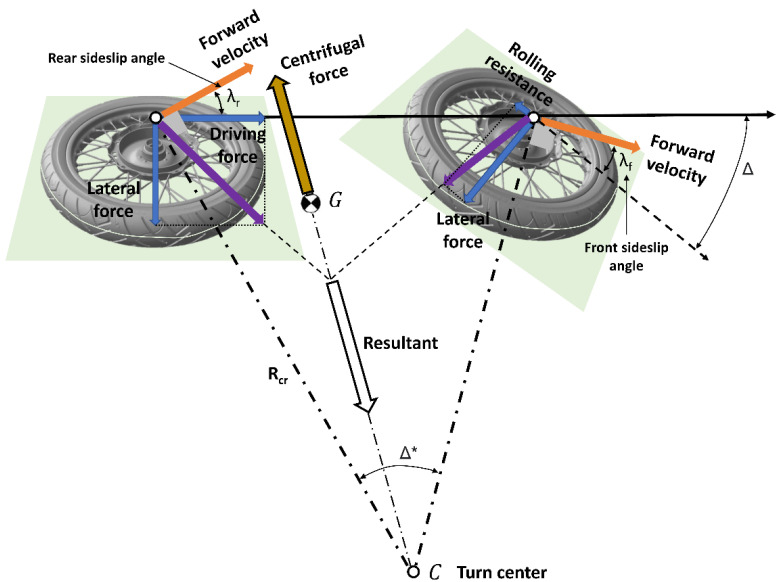
Forces acting on PTW vehicle during curvilinear trajectory with non-zero sideslip angle.

**Figure 2 sensors-23-01540-f002:**
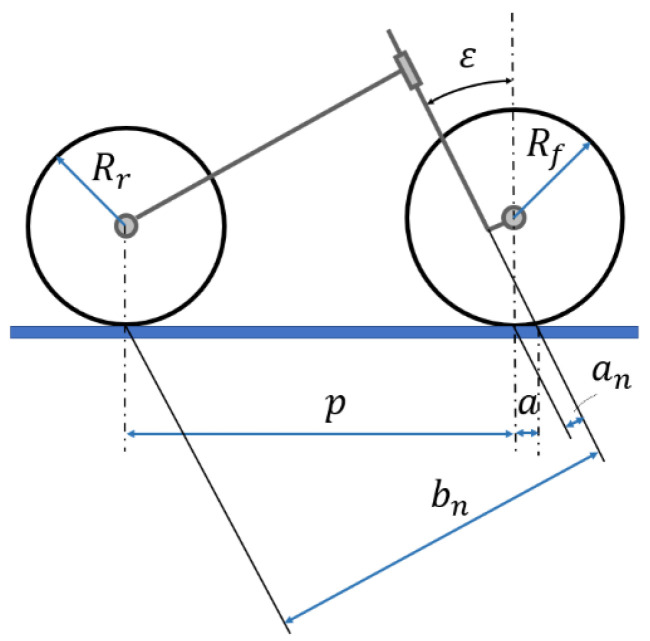
PTW main geometrical and inertial parameters.

**Figure 3 sensors-23-01540-f003:**
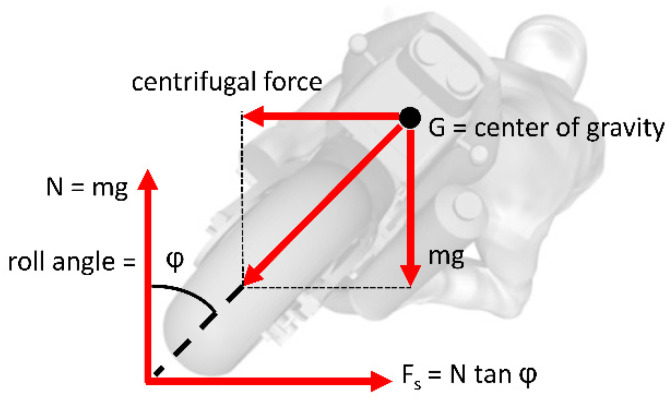
PTW lateral equilibrium during curvilinear trajectory.

**Figure 4 sensors-23-01540-f004:**
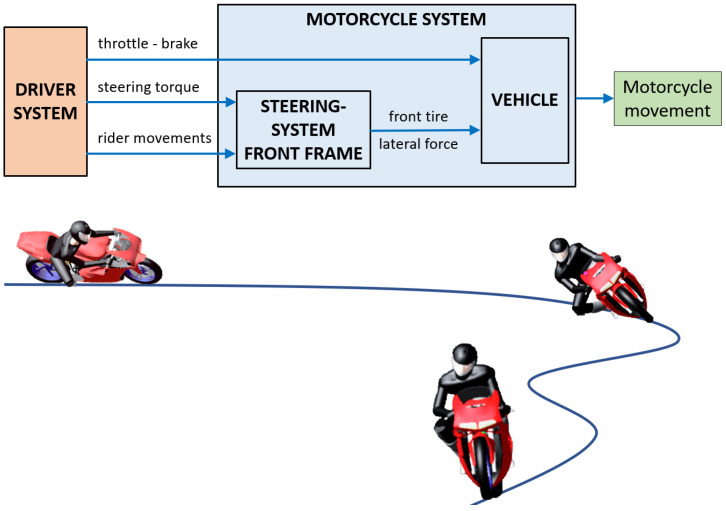
Scheme of the adopted PTW driver control system.

**Figure 5 sensors-23-01540-f005:**
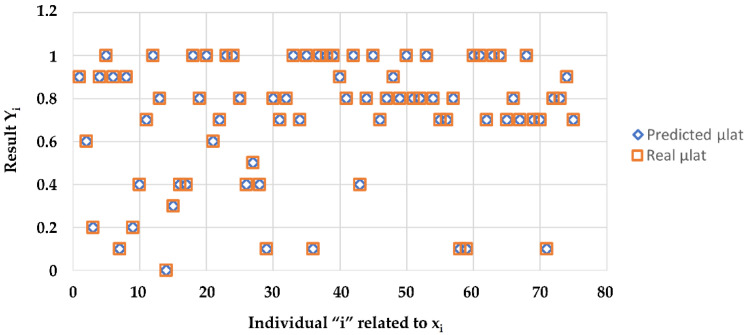
Coefficient *μ_lat_*: predicted v/s real values.

**Figure 6 sensors-23-01540-f006:**
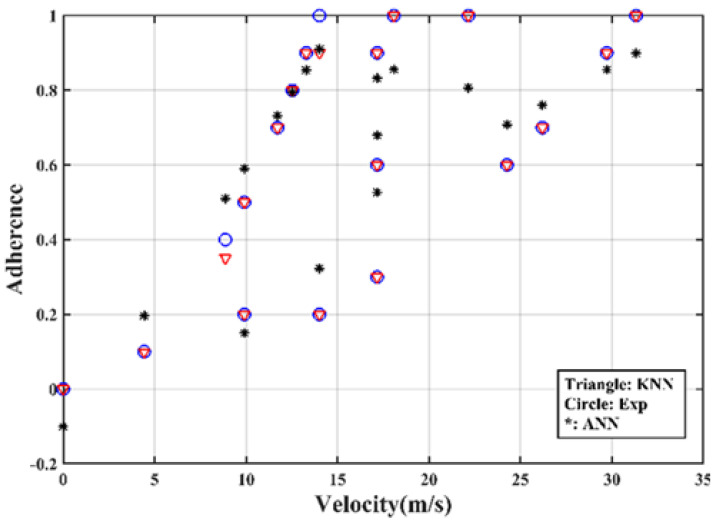
Adhesion values as a function of speed.

**Figure 7 sensors-23-01540-f007:**
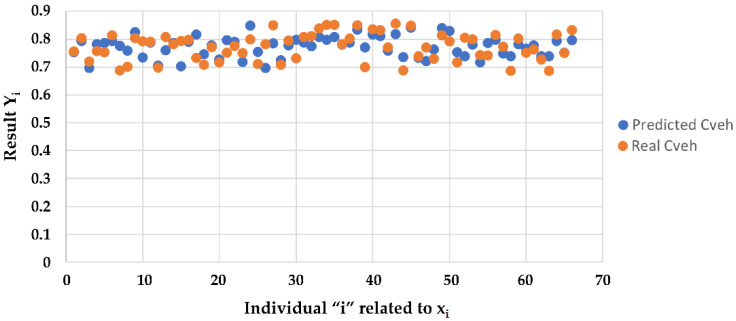
Coefficient *C_Veh_*: predicted v/s real values.

**Figure 8 sensors-23-01540-f008:**
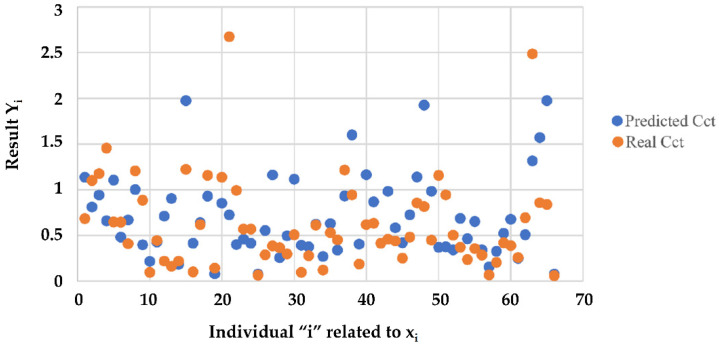
Coefficient *C_ct_*: prediction results.

**Figure 9 sensors-23-01540-f009:**
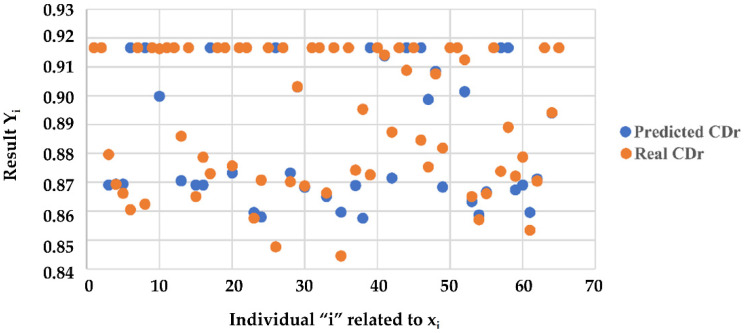
Coefficient *C_Dr_*: prediction results.

**Figure 10 sensors-23-01540-f010:**
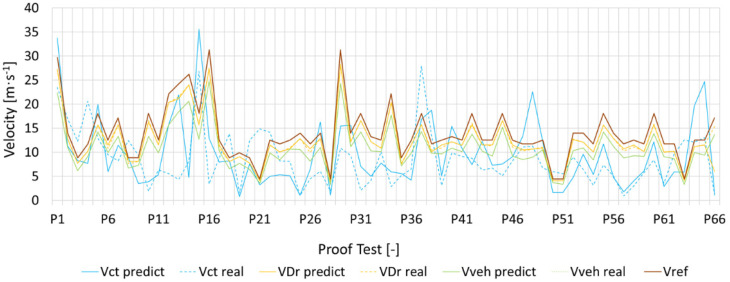
Predicted velocity vs. referent and real velocity.

**Figure 11 sensors-23-01540-f011:**
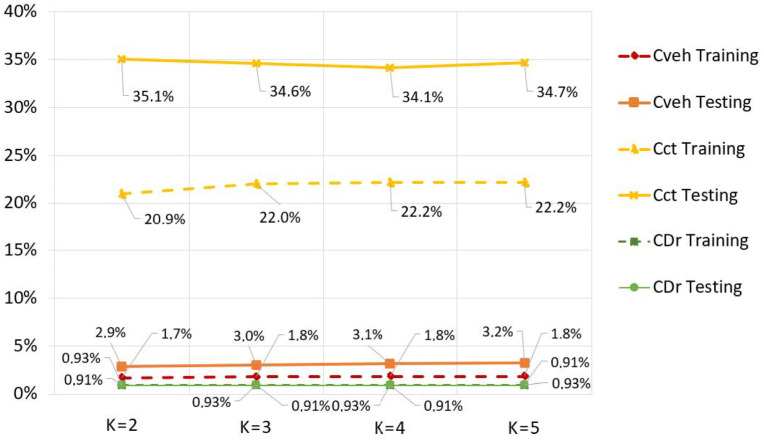
Evolution of the MAE indicator as a function of KNN.

**Figure 12 sensors-23-01540-f012:**
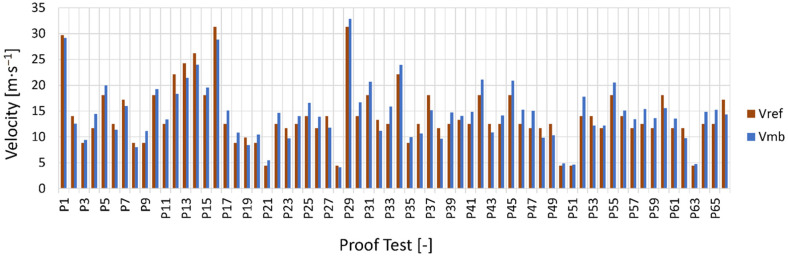
Comparison between maximum cornering velocities predicted with the present ML-KNN method and the calculated multibody model of PTW vehicle.

**Figure 13 sensors-23-01540-f013:**
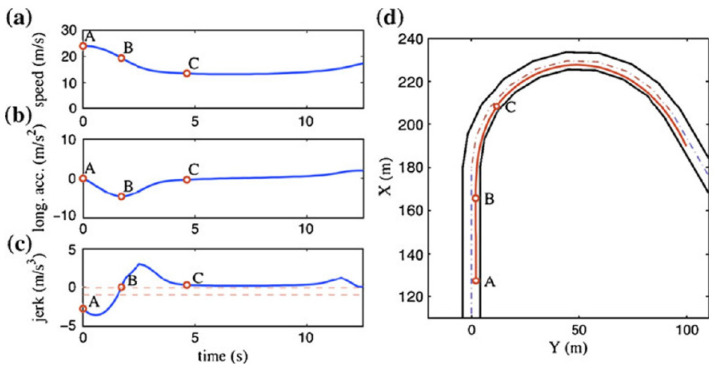
Recommended cinematic quantities in curvilinear trajectory. (**a**) optimal speed profile; (**b**) optimal longitudinal acceleration profile; (**c**) optimal jerk profile; (**d**) optimal trajectory.

**Table 1 sensors-23-01540-t001:** Principle, main advantage and disadvantage of existing methods.

Existing Methods	Principle	Advantages	Disadvantages/Limitations
Learning a Curve Guardian for Motorcycles [14].	Using IA to improve the curve undertaking. Analyzing the instruments’ measured data.	Lane localization, adding a learned roll prediction approach, using standard maps for real-world evaluation.	Does not anticipate the danger before undertaking the curve. Does not consider road grip in the estimation of roll angle.
Powered Two-Wheeler Riding Pattern Recognition Using a Machine-Learning Framework [15].	Using IA to predict rider behavior. Analyzing the instruments’ measured data.	Recognition of the driver action by applying ML on a dataset of measurement collected by sensors.	Does not reply completely to the challenge of anticipating dangers on the curve. Uses a set of sensors that may raise costs.
Powered Two-Wheelers Critical Events Detection and Recognition Using Data-Driven Approaches [16].	Development of critical event detection methodology by using AI classification algorithms.	The classification of the events with ML techniques, which could constitute a good database to characterize critical events.	The experiences collected based on a single model HONDA CBF 1000 which is not beneficial in case of generalization.
Estimation of Mental Workload during Motorcycle Operation [17].	Development of a method for mental workload when riding a PTW.	Contributes to characterize the driver behaviour by estimating the level of fatigue.	Deal partially with the challenges of PTW concerning curve undertaking.
Lateral & Steering Dynamics Estimation for Single Track Vehicle: Experimental Tests [18].	Development of on-board instrumentations for lateral and steering dynamic estimation.	Deals with lateral and steering dynamics estimation the reconstruction of unknown inputs.	Does not anticipate the danger before undertaking the curve. The loss of information due to linearization is considerable.

**Table 2 sensors-23-01540-t002:** Parameters used in algorithms for cornering assistance system.

Category	Parameter	Measure	Calculation/Estimation Method	Training Ranges
Characteristics of the vehicle	$R_{n}$	Geometric ratio	Provided (constant value)	[0.008; 0.061]
	$R_{f}$	Radius of front wheel [m]	Provided (constant value)	[0.085; 0.331]
	$R_{r}$	Radius of rear wheel [m]	Provided (constant value)	[0.084; 0.323]
Contact road-tire	$Ml_{Tire}$	Mileage record of the tire [Km]	Expressed as a ratio reflecting the state of wear of the wheel in relation to the mileage	[0.007; 0.995]
	$\mu_{lat}$	Friction coefficient of the Road	Estimated with ML method [7]	[0.003; 0.998]
	$P_{Tire}$	Pressure [Bar]	Expressed as a ratio reflecting the state of pressure (date < 3 months)	[0.5; 1.4]
	$Tire_{Type}$	Brand–width–Flank height–Tire structure–Diameter -load index–velocity index–Wet grip index [coding]	Provided (constant value codified as a decimal)	[0.003; 0.999]
Driver behavior	Time	Month/Day/Hour [coding]	Available information, Expressed as a ratio of Driving Time: Driver/Referent driver	[0.7; 0.99]
	$Ml_{driver}$	Record of the driver [Km]	Expressed as a ratio of mileage: Driver/Referent driver	[0.5; 0.99]
Referent vehicle	$c_{position}$	Radius (m)	Provided by GPS position	[0.01; 0.05]
	T	Temperature (°C)	Available information	Not considered
	H	Opto-electronic signal (mA)	Available information	Not considered
	Vis	weather visibility index	Available information	Not considered

**Table 3 sensors-23-01540-t003:** Parameter values used in the tests.

Test	Vehicle	Characteristics of the Vehicle			Contact Road-Tire				Driver Behaviour				Referent Vehicle
		*R_n_*	*R_f_* (m)	*R_r_* (m)	*Ml_Tire_* (ratio)	* μ_lat_ *	*P_Tire_* (ratio)	*Tire_Type_* (ratio)	*C_veh_*	*C_ct_*	Time (ratio)	Ml_driver_ (ratio)	*C_position_* (1/m)
P1	HONDA CBR500R/F/X	0.017	0.182	0.201	0.29	0.80	1.0343	0.56	0.7558	0.6817	0.7	0.5	0.01
P2	LAGENDA 115 R6	0.012	0.180	0.178	0.25	0.90	0.5000	0.76	0.8044	1.0988	0.7	0.5	0.02
P3	LAGENDA 115 FZ150	0.019	0.117	0.121	0.42	0.84	0.5000	0.94	0.7200	1.1731	0.7	0.7	0.02
P4	HONDA CBR500R/F/X	0.017	0.180	0.195	0.05	0.44	1.1060	0.26	0.7568	1.4538	0.7	0.8	0.01
P5	HONDA CB650F	0.016	0.202	0.218	0.22	0.57	0.5000	0.26	0.7534	0.6444	0.9	0.8	0.03
P6	LAGENDA 115 XJ6	0.012	0.205	0.212	0.31	0.71	1.2020	0.64	0.8141	0.6418	0.9	0.5	0.05
P7	LAGENDA 115 XV950R BOLT	0.021	0.254	0.239	0.24	0.39	0.5000	0.28	0.6882	0.4077	0.7	0.5	0.05
P8	HONDA CB650F	0.019	0.207	0.210	0.46	0.21	0.5000	0.28	0.7011	1.2042	0.9	0.5	0.01
P9	LAGENDA 115 R1	0.012	0.190	0.190	0.77	0.44	0.5000	0.71	0.8040	0.8822	0.7	0.5	0.01
P10	LAGENDA 115 NMAX 155	0.012	0.125	0.117	0.90	0.56	0.6346	0.91	0.7924	0.0945	0.7	0.5	0.05

## Data Availability

The corresponding authors will be happy to provide data that are not directly available in the article.

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
