# Peer review of "An Advanced Rider-Cornering-Assistance System for PTW Vehicles Developed Using ML KNN Method"

_sensors, 2023, doi:10.3390/s23031540_

Round 1
Reviewer 1 Report (New Reviewer)
The article is the product of a lot of work, but here are comments, suggestions and requested corrections:
Comments, suggestions and small errors of form:
Define the acronyms, or put a reference: What is ESC? What is KTM?
In [9] or [10] the name is misspelled (it is Cali or Calì ?)
Take "Below" - line 112. Put “next” for instance;
What is Gr? What is Gf? (front and rear wheel center of gravity¿), but must be quoted.
Line 206 – remove “above” put “previous” for example;
Line Define "rho"; defines "g" – equation (9);
line 241 - fix "precited";
line 243 "is the response..."
Line 252 - I had defined only CVe h...
Correct line 259 "precited"?
what is ci? Lowercase or uppercase?
line 262 - is the response ...
line 277 - is the response ...
The notation (font) of mathematical symbols, when outside the equation, should be in italics;
Make uniform notation. CDriver or Cdr (as in Fig. 10)?
Use uniform notation for equation ... equation 10;
K from capital KNN always?
Line 367 – standardize the call of equations...;
Content considerations:
Fig. 5 - prediction is exactly equal to the real value? Comment...;
Fig. 10 does not show that the result K=2 is the best for all algorithms... In fact, there is little difference between the values of K. It would not be the case to say that, depending on the case, in which an algorithm is better, would the corresponding K be chosen? For example, if the objective was to test the driver, then what would be the best K?
You could test the Kalman filter, even linear, there is already a version with parameter estimation (covariances) in real time. There are also non-linear filters (Extended and Unscented). Kalman Filter tests for this dynamic performance function should at least be cited in future work.
Author Response
Authors gratefully acknowledge the constructive comments on our manuscript. In the revised version, the amendments to the paper are highlighted using yellow colour. The authors have also better arranged the formatting of the manuscript for some inaccuracies in form and grammar. Now the manuscript was deeply modified according to the reviewers’ comments.
Questions |
Answers |
1 Define the acronyms, or put a reference: What is ESC? What is KTM? |
Authors thank the reviewers. Now the meaning of ESC, KTM and all acronyms is reported in the text of the Manuscript. |
2 In [9] or [10] the name is misspelled (it is Cali or Calì ?) |
Authors thank the reviewers. Now the correct surname “Calì” is reported in the text of the Manuscript. |
3 Take "Below" - line 112. Put “next” for instance; |
Authors thank the reviewers. Now the sentence in line 112 starts with the word “next”. |
4 What is Gr? What is Gf? (front and rear wheel center of gravity¿), but must be quoted. |
Yes, authors confirm that Gr and Gf are the centers of gravity of the front and rear wheels (line 131-132). |
5 Line 206 – remove “above” put “previous” for example; |
Authors thank the reviewer. Now the word has been changed as indicated. |
6 Line Define "rho"; defines "g" – equation (9); |
I µlat, ρ, vx, Ï•r , g are respectively: coefficient of static friction, the curvature of the turn, the longitudinal speed, the tilt angle of the road, gravity. See line 280-281. |
7 line 241 - fix "precited"; |
Now this sentence has been changed. The correct word is predicted (estimated). |
8 line 243 "is the response..." |
Now this sentence has been changed as suggested. |
9 Line 252 - I had defined only CVe h... |
All coefficients introduced in the study were defined in Section 3. |
10 Correct line 259 "precited"? |
Now the sentence has been changed. The correct word is predicted (estimated). |
11 what is ci? Lowercase or uppercase? |
Authors thank the reviewer. Now “ci” has been replaced with the (lowercase) curve indexed according to the measure à position = i (See table 2). |
12 line 262 - is the response ... |
Authors thank the reviewer. Now the sentence has been changed as indicated. |
13 line 277 - is the response ... |
Authors thank the reviewer. Now the sentence has been changed as indicated. |
14 The notation (font) of mathematical symbols, when outside the equation, should be in italics; |
Authors thank the reviewer. Now the mathematical symbols have been changed as indicated. |
15 Make uniform notation. CDriver or Cdr (as in Fig. 10)? |
Authors thank the reviewer. Now uniform notation for CDr has been used in the text. |
16 Use uniform notation for equation ... equation 10; |
Authors thank the reviewer. Now the mathematical symbols have been changed as indicated. |
17 K from capital KNN always? |
Yes, in our manuscript KNN can always be written with K capital. |
18 Line 367 – standardize the call of equations...; |
Authors thank the reviewer. Now call of equations were standardized. |
19 Fig. 5 - prediction is exactly equal to the real value? Comment...; |
As explained in the paragraph, the prediction of adherence coefficient was the subject of the authors previous paper. We meant to repeat the learning so that the new algorithm considers the theoretical values. Hence the similarity between predicted and real values.
We found that the predicted road grip given by the KNN model is the closest one to the experimental values. More than that, ANN is instable because of its probabilistic behaviour. By calculating R² indicator, we found that the ANN algorithm gives different values that variate from 25% to 90%. See line 290-306. |
20 Fig. 10 does not show that the result K=2 is the best for all algorithms... In fact, there is little difference between the values of K. It would not be the case to say that, depending on the case, in which an algorithm is better, would the corresponding K be chosen? For example, if the objective was to test the driver, then what would be the best K? |
In fact, the performance of the algorithms doesn’t behave the same way with the variation of K. Nevertheless, the only case which reacts significantly to the variation of K is that of Cct prediction. On the other hand, to optimize the algorithm resources we preferred to go for the least constraining. |
21 You could test the Kalman filter, even linear, there is already a version with parameter estimation (covariances) in real time. There are also non-linear filters (Extended and Unscented). Kalman Filter tests for this dynamic performance function should at least be cited in future work. |
Thank you for your suggestion, it will be a considerable added value to work with the filters (specially KALMAN). I believe that it could be complementary for our work on the deployment phase. Although given that the spirit of our project is to deploy less expensive instruments and to exploit more virtual estimators, hence our orientation towards ML technics. |

Reviewer 2 Report (New Reviewer)
This paper presents a reasonable method to solve a real application problem. It is well-organized, clearly writing, and shows some interesting results that encouraged to be accepted with major revision. However, the commented questions needs only to be answered.
1. Please explicitly indicate and clarify the challenges this study aims to address. What are the challenges and why? Why cannot the previous studies well address these challenges.
2. At the end of section 1 add a table that summarizes the advantages and disadvantages of existing methods facing the same problem. This way the reader would rapidly appreciate novelty of the paper.
3. The introduction and related work are mixed into a long section. It would be more clearly for readers to separate them
4. Please enrich the captions of all figures and tables for clarification.
5. Figure 4 needs more explanation.
6. Why do you build on the K-Nearest Neighbor (KNN) Machine Learning technique ? There are many state of the arts Machine Learning techniques that used with more popular and demonstrate better performance than KNN.
7. As shown in Figure 10, what is your explanation that the best result at K=2 for the KNN?
8. In the comparison to SOTA methods, more experimental results of other state-of-the-art methods should be given.
9. I also find some grammar problems in this paper. Author needs to carefully check these low mistakes, which is very important for readers.
Author Response
Authors gratefully acknowledge the constructive comments on manuscript. In the revised version, the amendments to the paper are highlighted using yellow colour. The authors have also better arranged the formatting of the manuscript for some inaccuracies in form and grammar. Now the manuscript was deeply modified according to the reviewers’ comments.
Questions |
Answers |
1 Please explicitly indicate and clarify the challenges this study aims to address. What are the challenges and why? Why cannot the previous studies well address these challenges. |
The challenges this study aims to address are: - The complexity of the PTW (Powered Two Wheelers) dynamics; - The lack of driver assistance systems for this type of vehicle; - The real-time dynamic parameters sharing of information; - The lack of databases in this field; These challenges are the main obstacles that obstruct the constitution of a development framework based on experience feedback and design an affordable driver assistance system for the majority of users of this type of vehicle. The majority of studies that addressed this topic (curve assistance system) used an instrumentational approach based on sensors or automatic estimators, but we could cite the previous studies conducted in the same spirit (Ferhat Attal. “Classification de situations de conduite et détection des événements critiques d’un deux roues motorisé”. Mathématiques générales [math.GM]. Université Paris-Est. Français. NNT :2015PESC1003. (2015)) in which the author aimed to characterize critical curve situations using AI classification algorithms. Indeed, our work complements his study. On the other hand, at this stage of the state of the art, the framework not yet being completely finalized, the works which arise complement each other. Notably the use of image recognition and AI for visual detection by camera, which is the main field analyzed by researchers currently. See line 81-109. |
2 At the end of section 1 add a table that summarizes the advantages and disadvantages of existing methods facing the same problem. This way the reader would rapidly appreciate novelty of the paper. |
Thank you for your suggestion. A table that summarizes the advantages and disadvantages of the existing methods facing the same problem has been added at the end of section 1. |
3 The introduction and related work are mixed into a long section. It would be more clearly for readers to separate them. |
Authors thank the reviewer. Now a new sub-section was added in the Introduction. |
4 Please enrich the captions of all figures and tables for clarification. |
Authors thank the reviewer. Now the captions of all figures and tables have been enriched. |
5 Figure 4 needs more explanation. |
Authors thank the reviewer. Now contents of Figure 4 have been explained more in detail. |
6 Why do you build on the K-Nearest Neighbor (KNN) Machine Learning technique? There are many state of the arts Machine Learning techniques that used with more popular and demonstrate better performance than KNN. |
Our Choice is for KNN for several reasons, namely: · As explained in section 3.1 (methodology description), given the nature of outputs (numerical observations) we chose regression form of KNN. · We conducted a comparison of the different methods and we found that the ANN and KNN gives the best results, but the ANN is unstable given its probabilistic behavior. By calculating R² indicator, we found that the ANN algorithm gives different values that deviate from 25% to 90%. · it is optimal, uses less resources and gives good performance, simple to deploy and to interpret. See line 81-109. |
7 As shown in Figure 10, what is your explanation that the best result at K=2 for the KNN? |
There is a slight difference in the performance of the algorithms with the variation of K (numbers of nearest neighbors), but we could observe a best performance with K=2 especially in the case of Cct prediction. This is explained by our database layout: the KNN algorithm exploits the correlation between the values, thus the results are limited to the extent of the database that we have used and the diversity of scenarios that it represents. See line 376-381. |
8 In the comparison to SOTA methods, more experimental results of other state-of-the-art methods should be given. |
As explained before (question 6), The SOTA comparison was already performed on our previous paper ([26] Fakhreddine, J., Bekkay, H. and Abderrahim, M. Controlling Powered Two-Wheeled vehicles in bends using machine learning. International Conference on Electronic Engineering and Renewable Energy Systems (ICEERE'22), Saidia, Morocco, 2022.), but considering the characteristics of our dataset and the advantages offered by KNN, we preferred to use this technique.
|
9 I also find some grammar problems in this paper. Author needs to carefully check these low mistakes, which is very important for readers. |
Authors thank the reviewers. The article has been reviewed by a native English speaker and all grammar errors in it have been corrected. |

Reviewer 3 Report (New Reviewer)
I appreciate your idea to estimate an appropriate value of the heeling angle of PTW by developing algorithms based on on-board measurements and agree that the sensors available to measure this variable are too bulky and expen-sive. Based on experience with sensor systems for autonomous vehicles I am afraid that single sensor(s) of one type might be insufficient to provide accurate information. The developed model based on ML and selected ML techniques appear to be promising although arguments for your choice (ANN, KNN) should be given. The parameters for the equilibrium control quantities are well defined and the accuracies of correction coefficients (the clue of your approach) well deduced and discussed . I took from the long list of references that a rigorous state-of-the-art assessment was made though most of the cited articles are surprisingly old. Finally, please clear the text for some minor faults and improve the clarity of Fig. 5 before publication.
Author Response
Authors gratefully acknowledge the constructive comments on manuscript. In the revised version, the amendments to the paper are highlighted using yellow colour. The authors have also better arranged the formatting of the manuscript for some inaccuracies in form and grammar. Now the manuscript was deeply modified according to the reviewers’ comments.
Questions |
Answers |
1 I appreciate your idea to estimate an appropriate value of the heeling angle of PTW by developing algorithms based on on-board measurements and agree that the sensors available to measure this variable are too bulky and expensive. Based on experience with sensor systems for autonomous vehicles I am afraid that single sensor(s) of one type might be insufficient to provide accurate information. The developed model based on ML and selected ML techniques appear to be promising although arguments for your choice (ANN, KNN) should be given. The parameters for the equilibrium control quantities are well defined and the accuracies of correction coefficients (the clue of your approach) well deduced and discussed. I took from the long list of references that a rigorous state-of-the-art assessment was made though most of the cited articles are surprisingly old. Finally, please clear the text for some minor faults and improve the clarity of Fig. 5 before publication. |
The authors express their gratitude to the reviewer. |

Round 2
Reviewer 2 Report (New Reviewer)
The manuscript is well-revised, and it is acceptable in its current form.
This manuscript is a resubmission of an earlier submission. The following is a list of the peer review reports and author responses from that submission.
Round 1
Reviewer 1 Report
This article is not like a standard academic paper at all. It is recommended to carefully study the standardized writing of academic papers.
Reviewer 2 Report
The paper presents an ML based approach for developing an advaced rider-cornering-assistance system for two-wheelers. The topic is interesting and of large relevance. However, due to many typos, strange styles (even slang), wrong usage of tenses, incomplete, unexplained and bad quality figures and graphics (e.g. missing axes labels), missing or "non-updated" literature sources, I have the impression of an incomplete article, which should be revised carefully by the authors themselves and an English native speaker (and not by the reviewer!) and corrected properly. I have no problem with the content of the paper, although I think, it needs also more professional input and professional handling, proof, intensive reading and correction. To my mind it could be an article for a nice conference, but has not the quality for a journal paper. To me, the work done is part of a bigger thing, which the authors could take into account, to prepare a sound journal article.
general remarks
- all figures should have better quality, some of them (e.g. 3) are almost unreadable
- all figures should be explained
The abstract should be shorter and contain only relevant information on how the paper contributes. To my mind almost the first half of the abstract can be dropped.
In the introduction the authors should update their numbers on the decrease of serious injuries in source [2]. What does the statement in line 63 mean? I don't understand!
Section 2 starts with an almost complete copy of the the previous paragraph of section 1. Further, a description of fig.1, i.e. what is happening while cornering, could help. If literature is available, please relate to it.
Not all parameters in paragraph 2.1 asscociated with figures 1, 2 and 3 are described.
No literature sources are given for figures 1 and 3 - at least figure 1 is availabe in source [9]. The authors should check all their figures and add literature sources.
The subsections 2.1-2.3 can be shortened by far. To my mind, it is clear that there are many parameters that are affected by "risk factors" like friction, the PTW rider, etc. Here, I suggest to start the description with a mathematical formulation (as done in 2.4) of the WHOLE(!) problem as a function of several factors, components, phenomenons, physical context, etc. (as shown in eq. (2)-(7)). Then, table 1 makes sense! As a result, these subsections can be shorter by far! What is the difference between eq. (2) and eq. (7)?
What does the attribute "correction" mean in line 202? Please explain!
Why does section 3 have this name? Further, the parameter/variable Vmax occurs only here and is never used again!
Figures 5 not helpful without any relevant information on the track, such es angles, steepness, inclinations angles, track length, etc. Otherwise it can be dropped.
There are redundant information in section 3.2.
Where does eq. (9) come from? At least, I'd expect a source and/or an explanation.
Figures 6ff have no axis description! Which blue dot corresponds to which orange? The reader is left alone here and might think abot the results by his/herself. All figures - if they are not immediately clear in a sudden - should be explained carefully! You can do that here, or in a separate section. The Discussion section could help, since there some interpretations are already given.
Figure 6 is difficult to understand, here, the authors should change the colors and/or forms. Why the assumption in lines 248 and 264 can be made? Further, the authors should change figures 111 and 122!
The text below 3.4 is almost the same as the text below 3.3!
Why is there an upper border near 0.92 in figure 9?
Why did you decide to use MAE? Why not RMS or other error metrics?
Does "K" in line 304 and figute 111 stands the K in KNN?
The authors don't spend a word on the test and the trainings data sets, e.g. how big are they? How many situations/cases/drives?
In line 338 the authors cite themselves in a not usual way.
The authors close the ariticle with a discussion. However, although the authors state and discuss their results, I interprete the method as "working", but neither superior to other methods. To my mind, the authors can resolve this, by carefully and soundly presenting their method, also by a decent comparison to the state of the art.
What is "IA" in line 352?
In line 374 a precision of 0.94% was reached. I guess, you mean an error?
Literature source [1] (line 390) has too few blanks. Also, the position of the initials are placed wrong.
The authors relate to their source [15], which I could not find searching the internet. If possible, I'd like to ask them to send it to me. Thank you.
Reviewer 3 Report
Medium level article.